# Scaled laboratory experiments explain the kink behaviour of the Crab Nebula jet

C.K. Li[1], P. Tzeferacos[2], D. Lamb[2], G. Gregori[3], P.A. Norreys[3], M.J. Rosenberg[1], R.K. Follett[4,5], D.H. Froula[4,5], M. Koenig[6,7], F.H. Seguin[1], J.A. Frenje[1], H.G. Rinderknecht[1], H. Sio[1], A.B. Zylstra[1], R.D. Petrasso[1], P.A. Amendt[8], H.S. Park[8], B.A. Remington[8], D.D. Ryutov[8], S.C. Wilks[8], R. Betti[4,5], A. Frank[4,5], S.X. Hu[4], T.C. Sangster[4], P. Hartigan[9], R.P. Drake[10], C.C. Kuranz[10], S.V. Lebedev[11] & N.C. Woolsey[12]

The remarkable discovery by the Chandra X-ray observatory that the Crab nebula's jet periodically changes direction provides a challenge to our understanding of astrophysical jet dynamics. It has been suggested that this phenomenon may be the consequence of magnetic fields and magnetohydrodynamic instabilities, but experimental demonstration in a controlled laboratory environment has remained elusive. Here we report experiments that use high-power lasers to create a plasma jet that can be directly compared with the Crab jet through well-defined physical scaling laws. The jet generates its own embedded toroidal magnetic fields; as it moves, plasma instabilities result in multiple deflections of the propagation direction, mimicking the kink behaviour of the Crab jet. The experiment is modelled with three-dimensional numerical simulations that show exactly how the instability develops and results in changes of direction of the jet.

[1] Plasma Science and Fusion Center, Massachusetts Institute of Technology, 77 Massachusetts Avenue, Cambridge, Massachusetts 02139 USA. [2] Department of Astronomy and Astrophysics, University of Chicago, 5640 South Ellis Avenue, Chicago, Illinois 60637, USA. [3] Department of Physics, University of Oxford, Parks Road, Oxford OX1 3PU, UK. [4] Laboratory for Laser Energetics, University of Rochester, Rochester, New York 14627, USA. [5] Department of Physics and Astronomy, University of Rochester, Rochester, New York 14627, USA. [6] LULI-CNRS, Ecole Polytechnique, CEA: Université Paris-Saclay; UPMC Univ Paris 06: Sorbonne Universités, F-91128 Palaiseau cedex, France. [7] Institute of Laser Engineering, Osaka University, Suita, Osaka 565-0871, Japan. [8] Lawrence Livermore National Laboratory, Livermore, California 94551, USA. [9] Department of Physics and Astronomy, Rice University 6100 S. Main, Houston, Texas 77521, USA. [10] Department of Atmospheric, Ocean and Space Science, University of Michigan, 2455 Hayward Street, Ann Arbor, Michigan 48103, USA. [11] The Blackett Laboratory, Imperial College London, London SW7 2BW, UK. [12] Department of Physics, University of York, York YO10 5D, UK. Correspondence and requests for materials should be addressed to C.K.L. (email: ckli@mit.edu).

X-ray images from the Chandra X-ray Observatory[1,2] show that the South-East jet in the Crab nebula changes direction every few years (Supplementary Fig. 1). This fascinating phenomenon is also seen in jets associated with pulsar wind nebulae and other astrophysical objects[3–5], and therefore is a fundamental feature of astrophysical jet evolution that needs to be understood[6–13]. The South-East Crab nebula jet is a highly collimated, mildly relativistic gas outflow from a pole of the rapidly rotating Crab pulsar, confined by a toroidal magnetic field (**B**) and accelerated outwards initially by means of magnetic fields (Poynting flux) drawing from the pulsar rotational energy[6,8–15].

Astrophysical jets can be studied in a controlled environment using appropriately scaled laboratory experiments that reproduce and study critical physical aspects; even though laboratory-generated supersonic plasma jets and astrophysical jets have very different scales, they can have similar dimensionless hydrodynamic and magnetic field parameters (as will be shown below) and therefore can share common physical properties[16–18]. These important similarities allow us to scale our laboratory results to the conditions in the Crab nebula, showing that the laboratory approach provides an incisive platform for studying various properties of astrophysical jets. To mimic the kink behaviour of the Crab jet, a laboratory experiment requires magnetic fields with the right properties: the fields must have a strong azimuthal (toroidal) component generated near the target where the jet is launched, and the fields must be embedded in ('frozen-in'), and advected with, the fast moving magnetized plasma flow.

The development and use of diagnostics that enable visualization and quantification of magnetic fields and magnetohydrodynamic (MHD) instabilities is as important as the creation of the plasma jet itself. Most conventional plasma diagnostics, using X rays and optical photons, are sensitive to plasma density and temperature but not to electromagnetic fields[19–21]. The recently developed method of monoenergetic proton radiography[22] (Methods section) is sensitive to electromagnetic fields and can provide spatial visualization and quantitative measurements.

Here we report experiments using scaled plasma jets, generated by high-power lasers, to reproduce and model the Crab jet (Methods section and Supplementary Fig. 2). Magnetic fields and current-driven MHD instabilities taking place in the jet, visualized and measured directly by monoenergetic proton radiography[22], have been unambiguously identified as the mechanisms that cause such a unique jet kink behaviour. We show how the toroidal magnetic field embedded in the supersonic jet triggers plasma instabilities and results in considerable deflections throughout the jet propagation, mimicking the kinks in the Crab jet. We also demonstrate that these kinks are stabilized by high jet velocity, consistent with the observation that instabilities alter the jet orientation but do not disrupt the overall jet structure. Our laser experiments produce plasma jets characterized by higher plasma temperatures ($> \sim$ keV) and faster flow velocities ($> \sim 1,000\,$km s$^{-1}$) that are at least one to two orders of magnitudes higher (faster) than those achievable by other experimental approaches[19–21]. Our experiments also produce plasma jets that have magnetic Reynolds numbers large enough for the magnetic field to be 'frozen into' the plasma flow. Consequently, the plasma in the jet must follow the field topology and its evolution, which is locally kinked but globally 'collimated' along the propagation axis. We successfully model these laboratory experiments with a validated three-dimensional (3D) numerical simulation, which in conjunction with the experiments provide compelling evidence that we have an accurate model of the most important physics behind the observed kinking of the Crab nebula jet. These experiments not only advance our knowledge of the structure and dynamics of the Crab jet, but also open up opportunities for laboratory study of jets from a variety of other astrophysical objects.

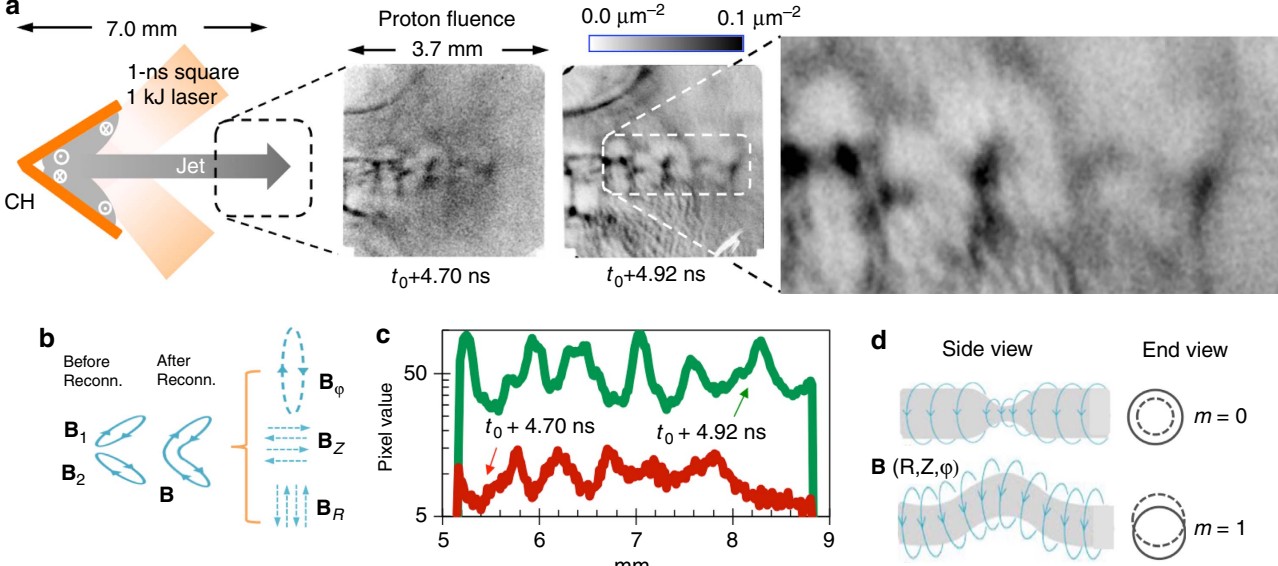

**Figure 1 | Experiments and proton radiographs. (a)** Schematic of a laser-beam-irradiated, cone-shaped target, and resulting plasma jet comprised of ions and electrons, also indicating the resulting toroidal magnetic field directions (Methods). Side-on (proton flux into the paper) radiographic images show the proton fluence distribution at $t_0 + 4.70$ ns with 15-MeV protons and at $t_0 + 4.92$ ns with 3.3-MeV protons, where $t_0$ is the time when the lasers turned on. The enlarged image shows a sequence of clumps and changes of jet direction. Cartoons in **b** illustrate the configurations of self-generated, spontaneous magnetic fields (**B**$_1$ and **B**$_2$) associated with the two plasma plumes. The resultant magnetic field can be decomposed into poloidal (**B**$_P$ = **B**$_R$ + **B**$_Z$) and toroidal (**B**$_\varphi$) components. The field structure is crucial for the excitation of the kinks. (**c**) Lineouts from the images in **a** along the axes of the plasma jets. The unit of the vertical axis is proton counts, which is proportional to the proton fluence. (**d**) Schematic illustrations of the fastest growing MHD current-driven instabilities: mode $m = 0$ (sausage, leading to jet propagation clumping) and $m = 1$ (kink, leading to jet direction changing). Higher modes ($m > 1$) are also expected to be excited, but will have smaller effects and are not illustrated here.

## Results

**Laser-driven scaled plasma jet.** In our laboratory experiments, we form plasma jets through the collision of expanding, laser-driven plasma plumes (Fig. 1). The laser-foil interaction induces self-generated magnetic fields due to the Biermann battery effects[23] ($\partial\mathbf{B}/\partial t \propto \nabla n_e \times \nabla T_e$, where $n_e$ and $T_e$ are electron density and temperature, respectively) that are predominantly toroidal with respect to each plume and to the jet that is formed from the collision of these plumes (Fig. 1b). The magnetic fields are embedded in and advance with the moving jet ($\partial\mathbf{B}/\partial t \propto \nabla \times (\mathbf{v}_j \times \mathbf{B})$, where $\mathbf{v}_j$ is jet velocity), mimicking the fundamental scenario that magnetic fields are anchored in the rapidly spinning Crab pulsar and advected with the Crab jet.

Figure 1a shows proton radiographs of the plasma jets at different times, indicating a structure that is collimated throughout its propagation but has a sequence of clumps and changes of direction along its length. These features reflect perturbations in the magnetic field structure around the jet (Fig. 2), and they grow locally and expand at each axial position where the jet is stable. The shape of the jet is serpentine due to the kink instability, and so can be viewed as comprised of ellipsoidal blobs that are typically viewed with the proton radiography from an angle $\theta \sim 45°$. Adopting this picture, we can apply the analysis of ref. 24 for ellipsoidal blobs. Interpreting the structures seen in the proton radiography images as caustics, we use the criterion for caustic formation[24], which indicates that $B > 0.8$ MG for $\theta = 45°$. Shown in Fig. 1d are cartoons illustrating the most feasible and fast growing MHD current-driven instabilities: Mode $m = 0$ (sausage) arises as the $\mathbf{B}_\varphi$ tension is enhanced by radial contraction, responsible for the axis pinching when $|\mathbf{B}_\varphi| > \sqrt{2}|\mathbf{B}_P|$. Mode $m = 1$ (kink) arises when the strength and pressure of $\mathbf{B}_\varphi$ increase at the inside of the kinks and decrease outside. $|\mathbf{B}_\varphi||\mathbf{B}_P|^{-1}\lambda(2\pi r_j)^{-1} > \alpha$, with $\alpha = 1$ is the Kruskal–Shafranov criterion for the kink instability[25], where $\lambda$ is the modulation wavelength and $r_j$ the jet radius. In astrophysical jets, effects like expansion can tend to stabilize the jet[10], resulting in $\alpha$ larger than but of order unity. This scenario is illustrated by experiment with a flat target (Fig. 3) where the plasma jet is stabilized when magnetic field is overwhelmed by the parallel components as the toroidal components around the jet are too weak to excite the MHD instability. The unstable modes have a growth rate $\gamma$ comparable to the inverse of the time required for phase velocity of an Alfvén wave ($v_A = B/\sqrt{4\pi\rho}$) to cross the unstable jet column[26]

$$\gamma = \Gamma\left(\frac{2\pi r_j}{\lambda}\right)\frac{v_A}{r_j}. \qquad (1)$$

Using the measured values $r_j \sim 0.5$ mm and $\lambda \sim 0.6$–0.7 mm, and $v_A \sim 1,000$ km s$^{-1}$ around the jet launching region, we find $\gamma \sim 3 \times 10^9$ s$^{-1}$, which is consistent with the instability evolution time implied by Fig. 1a.

**Modelling of experiments with numerical simulation.** To model the observations of the plasma jet, a 3D numerical simulation was performed with the radiation-MHD code FLASH[27,28] (Methods section). The simulation was post-processed to provide a more complete physical picture of the jet behaviour, leading to the images in Fig. 4 showing the spatial variations of various quantities at $t \approx t_0 + 5.0$ ns in the plane containing the jet's axis. Figure 4a shows that a modulated central 'spine' (backbone) region with stronger field strength is formed, and is surrounded by asymmetrically distributed, weaker fields around the jet core. When the field is sufficiently large and has nonuniform toroidal components $B_\varphi$, current-driven MHD kink modes are excited with the susceptibility increasing with increasing $|\mathbf{B}_\varphi/\mathbf{B}_P|$ (Fig. 4b). Such a structure is confirmed by the corresponding distribution of $\beta = 8\pi nkT/B^2$ (the ratio of plasma thermal to magnetic pressures) in Fig. 4c: in the jet core $\beta < \sim 1$, showing the flow is magnetically dominated, while in the surrounding plasma $\beta > 1$. This indicates that the jet is globally collimated due to inertial confinement and magnetic tension, but locally kinked.

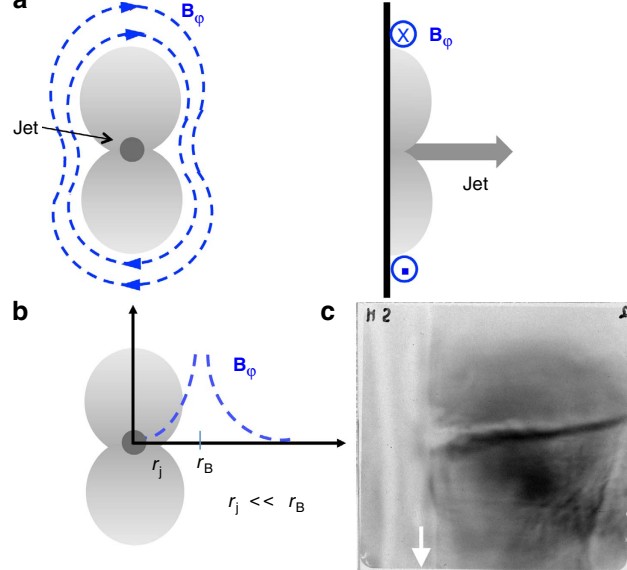

**a**

**b**

**c**

**Figure 3 | Plasma jet and magnetic field configurations generated by a laser-driven flat target.** (**a**) Cartoons of face-on and side-on views of a plasma jet and associated toroidal magnetic fields after reconnection due to collision of two plasma plumes from a laser-driven plastic foil. (**b**) Radial distribution of magnetic fields indicate the toroidal components around the jet are too weak to excite the MHD instability (overwhelmed by the parallel components, that is, $B_\varphi/B_P \ll 1$). (**c**) Proton side-on radiographic image shows the jet is stable to MHD instabilities when toroidal components are weak (white arrow points the position of flat foil target). It also suggests that in this type of experiment the toroidal fields generated by the plasma current are too weak to destabilize the jet propagation. The jet is predominately collimated by inertial confinement due to the hydrodynamic compression produced by the collision of the two plumes.

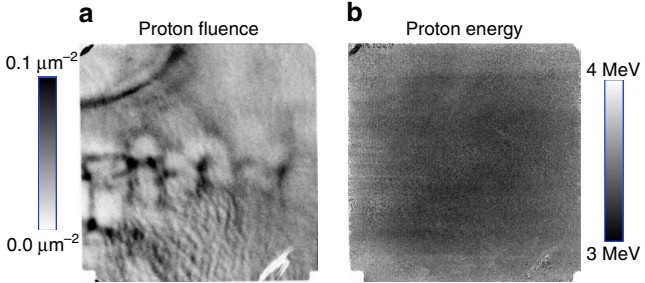

**a** Proton fluence

0.1 μm$^{-2}$

0.0 μm$^{-2}$

**b** Proton energy

4 MeV

3 MeV

**Figure 2 | Two images connected with each radiograph.** (**a**) Image of proton fluence versus position, taken with 3-MeV DD protons at 4.92 ns from the onset of the laser drive on the subject cone target, showing a clear kink structure which indicates that the jet propagation was subjected to plasma instabilities. (**b**) Image displaying mean proton energy versus position shows a very uniform distribution, with no hint of the density structure of the jet. The latter would be expected if Coulomb scattering[43] were important, indicating that the structures seen in the fluence image are due to deflections of protons by magnetic fields.

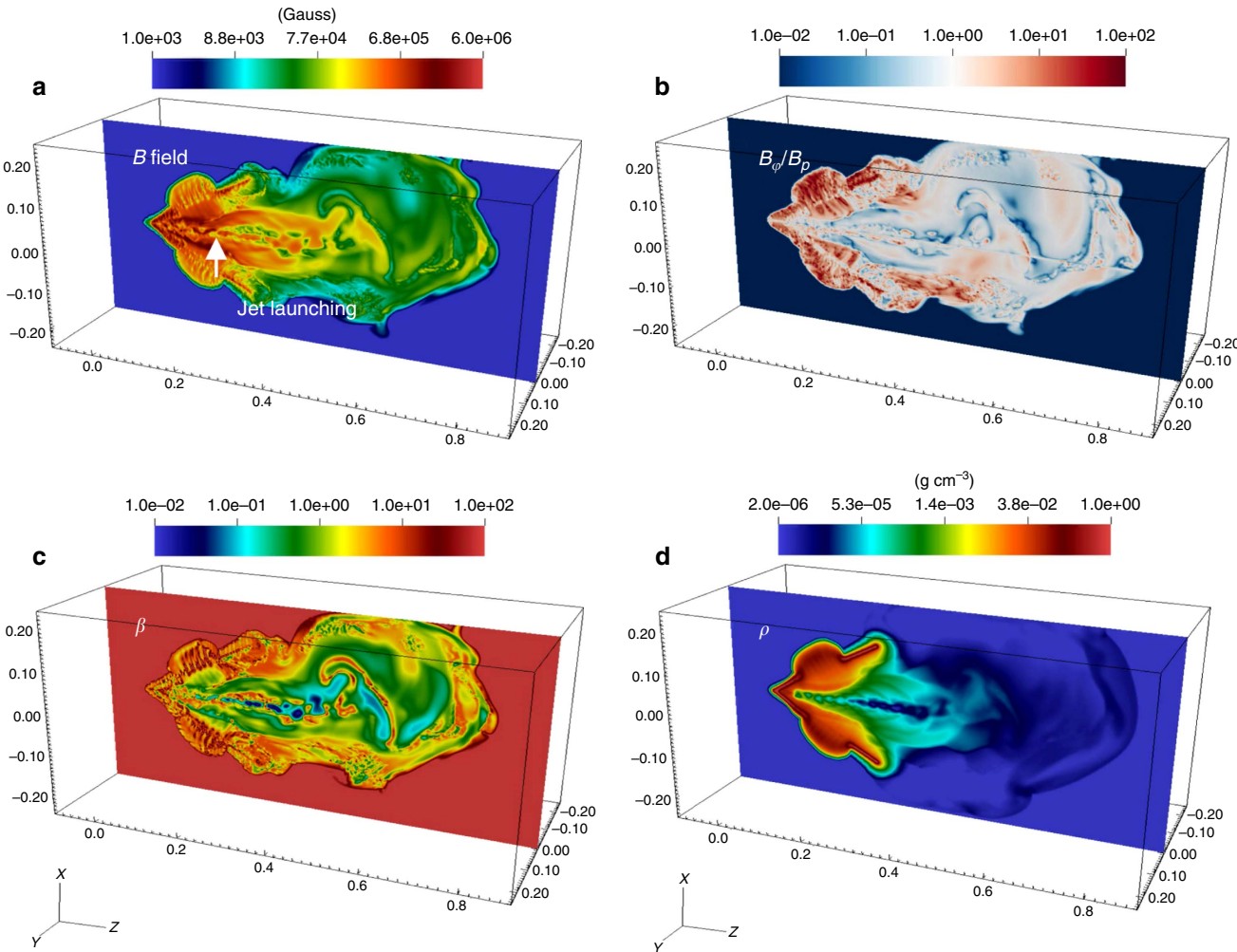

**Figure 4 | Images of various physical properties in the 3D numerical simulation of the jet.** (Images correspond to $t = t_0 + 5$ ns, the detailed evolution of the simulation can be accessed online in Supplementary Movies 1,2,3 and 4). (**a**) The amplitudes of the self-generated magnetic fields that are advected with the jet flow show a collimated flow with a wiggling central 'spine'. Outside the jet surface, the bulk flow has asymmetrically distributed magnetic fields. The white arrow indicates where the jet is thermally launched ($z \sim 2$ mm). (**b**) Image showing the ratio of toroidal ($B_t = |\mathbf{B_\varphi}|$) to poloidal ($B_p = |\mathbf{B_z + B_R}|$) field components. The image shows the locations where jet kinks take place and grow are correlated with the regions where $\mathbf{B_\varphi}$ is stronger and asymmetrically distributed. (**c**) The corresponding distribution of the ratio $\beta$ of plasma pressure to magnetic pressure. The jet core and surrounding region (bulk flow) have $\beta < 1$. The instability occurs in the region where advection of the magnetic field is dominant and $\beta < 1$. (**d**) The simulated plasma density distribution shows 'clumps' and 'kinks' corresponding to the field topology. (The units for x, y, and z axes are cm.)

The jet is not in force-free equilibrium: the gradient of thermal pressure is not in local balance with the sum of magnetic-pressure gradient and magnetic tension (hoop stress $-B_\varphi^2/4\pi r$); that is,

$$-\frac{\partial}{\partial r}\left(\frac{B_\varphi^2 + B_P^2}{8\pi}\right) - \frac{B_\varphi^2}{4\pi r} \neq \frac{\partial P}{\partial r}. \qquad (2)$$

The 'clumps' and 'kinks' shown in Fig. 4a–c are similar to those we observe in the experiments (Fig. 1a). The distribution of the simulated plasma density depicted in Fig. 4d shows a clear kink structure that is correlated with the field topology in Fig. 4a.

**Validation of numerical simulations.** The FLASH simulation was used to predict the physical properties of the jet (Methods section). These were compared quantitatively with the experimental measurements, including proton radiography (Fig. 1) and Thomson scattering[29] (Supplementary Fig. 3). Figure 5a shows the measured jet positions and velocities which match the simulations well, providing compelling evidence for the validity of the numerical simulation. The velocity at the front of

the jet after it has been traveling for several nanoseconds is estimated to be $v_j \sim 1,500$ km s$^{-1}$, indicating supersonic jet propagation with an internal Mach number $M \sim 3$. Such a high jet velocity has two important effects on jet propagation. First, high Mach numbers suppress the Kelvin-Helmholtz instability, lessening the entrainment of the surrounding plasma in the jet plasma. Second, the high jet velocity helps to move the 'frozen-in' non-uniform fields, leading to smoothing of asymmetric magnetic field line distributions, stabilizing the jet. Further validation is provided in Fig. 5b,c, where plasma densities and temperatures measured using Thomson scattering are plotted against the time-resolved jet positions, respectively (Supplementary Fig. 3), and agree well with the numerical simulation. Again, this consistency greatly increases our confidence that the simulation has captured the most important physics in the experiments.

## Discussion

The magnetization parameter ($= B^2/8\pi\rho v_j^2$, the ratio of the jet magnetic to ram pressures) shown in Fig. 6a is $\sigma \geq 1$ near the region where the jet was launched, and $\sim 10^{-2}$–$10^{-3}$ near the

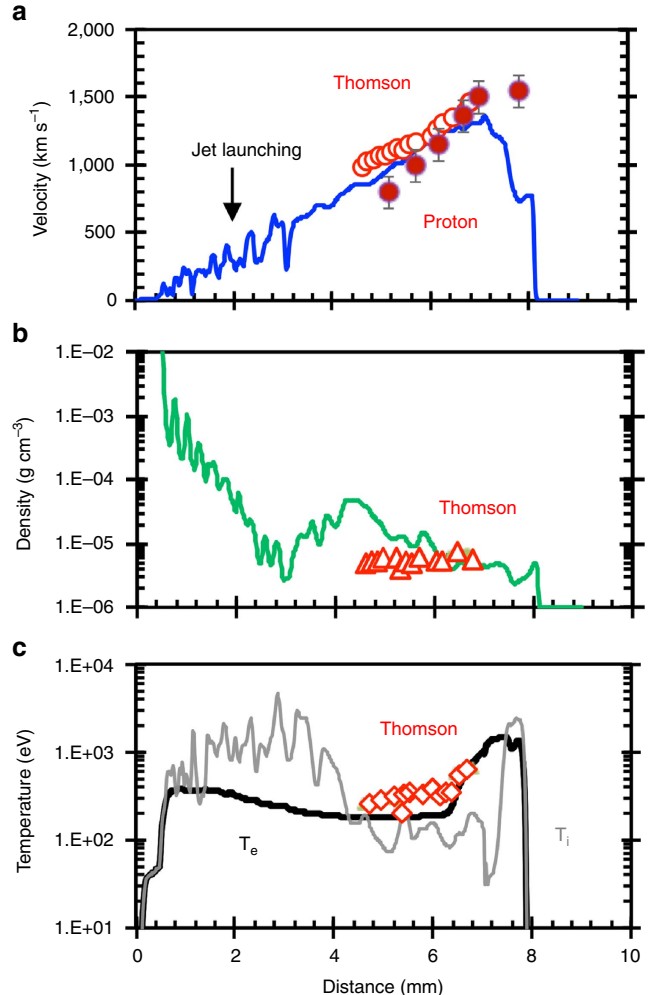

**Figure 5 | Comparison between measurements and numerical simulation.** (**a**) Measured jet velocities (solid circles by protons (Fig. 1c) and open circles by Thomson-scattering (Supplementary Fig. 3), with measurement uncertainties (error bars) discussed therein, respectively), plotted as a function of position in the jet flow compare well with simulated values (blue line). The error bars of proton measured jet velocities indicate $\Delta v \sim \pm 80\text{–}120 \text{ km s}^{-1}$, including measurement uncertainties and consequences of proton Coulomb scatterings. The increase in the simulated jet velocity as the flow propagates outwards, is a consequence of the gradient in the thermal plasma pressure, and leads to the decrease in the simulated jet density shown in (**b**) (green line). The measured plasma densities inferred from the Thomson-scattering data, which are shown as open red triangles, agree reasonably with those of the simulation. (**c**) The plasma temperatures T inferred from Thomson-scattering measurements (assuming $T_e \sim T_i$, in this relevant region, see Supplementary Fig. 3)[29,44], which are shown as open red diamonds, compare reasonably well with those of the simulation (black line).

jet head. These values compare very well to those of the Crab nebula, where observations and simulations indicate that $\sigma \geq 1$ close to the pulsar pole where the jet is launched, and $\sigma \sim 10^{-2}\text{–}10^{-3}$ near the termination shock where the jet becomes subsonic[13]. The morphological similarities between the Crab jet (Supplementary Fig. 1) and the laboratory jet can be clearly seen in the simulated current density map (Fig. 6b). The latter reveals kinks, knots, and large-scale radial deflections that are reminiscent of the structures and dynamics observed in the Crab pulsar outflow. This picture of a 'current-carrying' jet is in

agreement with existing numerical efforts on modelling the Crab jet[10,13] and the morphology mimics the jet structures observed in Chandra X-ray imaging[1].

These similarities provide rigorous justification of the relevance of the plasma jet to the Crab nebula jet, preserving the facts that the energy flux is predominantly carried by the Poynting flux close to the pulsar pole and by the particles close to the termination shock. The consistency between the experiments and the simulation provides compelling evidence that strong toroidal magnetic fields and the associated MHD kink instabilities are the cause of the observed jet structure, and that the simulation has captured the basic physics behind kink behaviour in jets. Furthermore, this comparison confirms the hypothesis that the observed directional change of the Crab jet can be caused by strong toroidal magnetic fields and associated MHD kink instabilities.

Other evidence of the relevance of our experiments to the jet in the Crab nebula is provided by several important dimensionless parameters. Both jets have a Lorentz factor of the order of unity ($\Gamma = 1$ for the laboratory plasma jet and $\Gamma \approx 1.09$ for the Crab Nebula jet[30]). Similarity in the MHD equations requires that the dissipative processes be negligible for both systems. This requirement is met if the viscosity, thermal conduction, and magnetic diffusion terms can be neglected in the momentum, energy, and generalized Ohm's law equations. Equivalently, a number of corresponding dimensionless numbers, such as the Reynolds number $Re(=Lv_j/\nu$, the ratio of inertial forces to viscous force, where $L$ is jet scale size and $\nu$ is the kinematic viscosity), the Péclet number $Pe(=Lv_j/\kappa$, the ratio of heat convection to conduction, where $\kappa$ is the thermal diffusivity), and the magnetic Reynolds number $R_{Me}(=Lv_j/D_m$, the ratio of flow velocity to diffusion velocity, where $D_m$ is the magnetic diffusivity) must be large in both systems. Table 1 shows that all of these numbers are large, demonstrating that these important conditions are met. Table 1 also lists the other physical parameters and dimensionless numbers that are relevant to this laboratory jet and to the jet in the Crab nebula. To scale the laboratory results to the environment of the Crab nebula, the MHD equations need to be invariant under the transformations given below for the two systems[17,18]:

$$r_{lab} = a r_{crab}; \qquad \rho_{lab} = b \rho_{crab}; \qquad P_{lab} = c P_{crab};$$
$$v_{lab} = \sqrt{\tfrac{c}{b}} v_{crab}; \qquad \tau_{lab} = a\sqrt{\tfrac{b}{c}} \tau_{crab}; \qquad \mathbf{B}_{lab} = \sqrt{c}\mathbf{B}_{crab}, \qquad (3)$$

where the subscripts 'lab' and 'crab' refer to the laboratory and Crab nebula jets, respectively. As shown in Table 1, excellent MHD scaling is obtained with $a \sim 1.6 \times 10^{-20}$, $b \sim 1.7 \times 10^{25}$ and $c \sim 1.1 \times 10^{19}$.

In summary, our scaled laboratory experiments and validated numerical simulation reveal that the change in direction observed in the Crab jet can be attributed to magnetic fields and the associated MHD kink instabilities. This work not only advances our knowledge of such jet structure and dynamics, but also opens up tremendous opportunities in the laboratory to explore jets from a variety of other astrophysical objects, including active galactic nuclei, young stellar objects, X-ray binary systems and pulsar wind nebulae.

## Methods

**Experiments.** In our experiment, performed at the OMEGA Laser Facility[31] and illustrated schematically in Supplementary Fig. 2, the plasma jet was generated by the interaction of laser beams with a special target. The target was constructed by two 50-μm-thick, $3 \times 3$ mm plastic (CH) foils separated by 60°. Each individual foil was driven by two laser beams (0.351 μm in wavelength) at an angle $\sim 28°$ to the foil normal, with total energy $\sim 1,000$ J in a 1-ns, square-top laser pulse with full spatial and temporal smoothing. The laser spot has a diameter of $\sim 850$ μm determined by phase plate SG4 (defined as 95% energy deposition), resulting in a laser intensity of order of $\sim 2 \times 10^{14}$ W cm$^{-2}$. Laser ablation generated a plasma

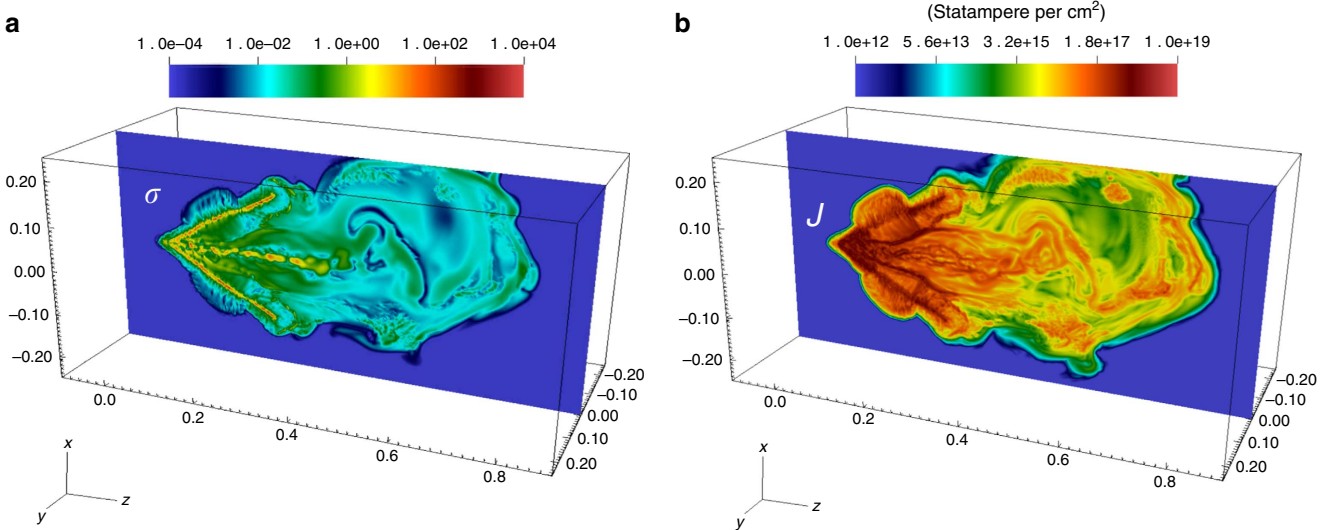

**Figure 6 | Spatial distributions of the magnetization parameter σ and the current density J.** (**a**) Simulated spatial distribution of the magnetization parameter σ at $t = t_0 + 5$ ns. (The detailed evolution of the simulation can be accessed online in Supplementary Movie 5). Near the region where the jet was launched ($z \sim 2$ mm), both the core of the jet and the bulk flow have $\sigma \geq 1$, while near the jet head $\sigma \sim 10^{-2}$–$10^{-3}$. (**b**) Simulated spatial distribution of the current density ($J$) at $t = t_0 + 5$ ns. The image clearly shows the kinked morphology of the jet. (The units for $x$, $y$, and $z$ axes are cm.)

### Table 1 | Physical parameters and similarity scaling between the laboratory jet and the Crab nebula jet.

| Parameters and scales | | Plasma jet in OMEGA experiment* | Scaled to the Crab nebula† | The kinked jet in the Crab nebula† |
|---|---|---|---|---|
| Temperature | $T_e$ | $\sim 300$ eV | | $\sim 1$–$130$ eV |
| Ionization state | $Z$ | $\sim 3.5$ | | $\sim 1$ |
| Number density | $n_e$ | $\sim 5 \times 10^{19}$ cm$^{-3}$ | | $\sim 10^{-2}$ cm$^{-3}$ |
| Pressure | $P$ | $\sim 4 \times 10^5$ bar | | $\sim 4 \times 10^{-14}$ bar |
| Jet radius | $r_j$ | $\sim 5 \times 10^{-2}$ cm | | $\sim 1$ pc |
| Jet velocity | $v_j$ | $\sim 400$ km s$^{-1}$ | $< 3 \times 10^5$ km s$^{-1}$ | $\sim 1.2 \times 10^5$ km s$^{-1}$ |
| Time scale | $\tau$ | $\sim 10^{-9}$ s | $\sim 1.5$ years | $\sim$ few years |
| Magnetic field | $B$ | $\sim 2$ MG | $\sim 0.6$ mG | $\sim 1$ mG |
| Thermal plasma beta | $\beta$ | $\sim 0.1$–$1$ | | $\ll 1$ |
| Magnetization parameter | $\sigma$ | $\sim 1$–$6$ | | $\geq 1$ |
| Mach number | $M$ | $\sim 3$ | | $\gg 1$ |
| Reynolds number | $Re$ | $\sim 2 \times 10^3$ | | $\sim 2 \times 10^{17}$ |
| Péclet number | $Pe$ | $\sim 1$–$5$ | | $\sim 4 \times 10^{15}$ |
| Magnetic Reynolds number | $Re_M$ | $\sim 3 \times 10^3$ | | $\sim 1 \times 10^{22}$ |
| Biermann number | $Bi$ | $\sim 6$ | | $\sim 6 \times 10^8$ |
| Radiation number | $\Pi$ | $\sim 3 \times 10^5$ | | $\sim 1 \times 10^{18}$ |

*Near the region of jet launching.
†Near the region of the pulsar pole.
The bold entries show the physical quantities from the two systems that can be directly compared through the scalings in equations (3), manifesting how the laboratory experiment parameters scale to match those of the Crab nebula jet.

plume on each foil, and the collision of these plumes forms a high Mach number plasma jet that propagates into the OMEGA chamber[32]. During laser illumination and heating, $\sim$ Megagauss $B$ fields (predominantly toroidal) are generated around each expanding, hemispherical plasma plume because of the Biermann battery effect[23] due to non-collinear electron density and temperature gradients ($\nabla n_e \times \nabla T_e$). The collision of the plasma plumes with $B$ fields of opposing sign eventually results in magnetic reconnection, leading to the formation a new magnetic topology with strong toroidal fields around the plasma jet[32].

**Proton radiography.** Monoenergetic proton radiography[22] has been developed on the OMEGA laser facility and utilized for backlighting of laser-produced plasma jets. From the Lorentz force ($\mathbf{F_L} = q(\mathbf{E} + \mathbf{v} \times \mathbf{B})$), deflections due to magnetic fields can be estimated as:

$$\xi = -\frac{q(A-a)a}{A m_\mathrm{p} V_\mathrm{p}} \int \mathbf{B} \times \mathrm{d}\boldsymbol{l} \qquad (4)$$

where $a (= 1$ cm$)$ and $A (= 28$ cm$)$ are distances from backlighter to the subject target and to the detector in this experiment, respectively; $m_\mathrm{p}$ is the proton mass and $V_\mathrm{p}$ is the proton velocity; $q$ is the proton electric charge, $\xi$ is the proton

deflection distance and d$\boldsymbol{l}$ is the differential pathlength along the proton trajectory. This technology[22] consists of a monoenergetic proton backlighter source and a matched imaging detector.

The backlighter is formed by an exploding-pusher implosion with a D$^3$He- (deuterium-helium-3) filled, glass-shell capsule[22] driven by 16–30 of the 60 OMEGA laser beams[31]. The capsule has a typical diameter $\sim 420$ μm and shell thickness $\sim 2$ μm, filled with 18 atm of equimolar D$^3$He gas. The laser delivered $\sim 10$ kJ in a 1 ns square pulse. Supplementary Table 1 summarizes the characteristics of the typical backlighter used in these experiments. The timing of the backlighter implosions was adjusted to provide radiographic images at different times relative to when the lasers turned on. The detection system[33] consists of a layered assembly of metallic foils and solid-state nuclear track detector CR-39 on which backlighting protons are recorded at 100% efficiency. The CR-39 has a chemical composition of $C_{12}H_{18}O_7$. When a charged particle passes through CR-39, it leaves a trail of damage along its track in the form of broken molecular chains and free radicals. The amount of local damage along the track is related to the local rate at which energy is lost by the particle. In particular, since $dE/dx$ is different for protons at different energies, protons with different energies result in different track diameters. In this experiment, the CR-39 is etched for 2–3 h in a 6N solution of NaOH, which reveals the tracks with diameters on the order of $\sim 10$ μm. An automated microscope system scans and records information about

the protons tracks, including their location on the piece of CR-39. Custom software is used to determine track properties and to transform that information into an image of proton fluence incident on the CR-39.

**3D numerical simulation.** The 3D Cartesian radiation-MHD simulation of the experiment was performed using FLASH[27,28,34], a publicly available, multi-physics, finite-volume, shock-capturing code[27]. The simulation takes advantage of the full range of HEDP capabilities of the code, so as to accurately model the physical processes in play. The MHD equations are evolved using a directionally unsplit staggered mesh solver[35], extended to three temperatures[36], adding also the Biermann battery effect[23,37,38]. We include non-ideal effects such as explicit Spitzer resistivity, implicit thermal conduction and heat exchange, as well as multi-group radiation diffusion with multi-material tabulated opacities and equations of state. The laser energy deposition is accurately modelled using a 3D optical ray trace laser package[39].

The computational domain spans 0.5 cm in X and Y, and 1 cm in Z (Supplementary Fig. 4), and is discretized on $\sim 3.3 \times 10^7$ zones ($\sim 20\,\mu m$ cell size). The reconstruction is carried out with a Piecewise Parabolic Method[40], employing a minmod limiter. The Godunov fluxes are recovered with an HLLC (Harten, Lax and van Leer-Contact)[41] Riemann solver. Outflow boundary conditions are imposed on all sides. The experimental target is modelled as two $3 \times 3\,mm$ polystyrene foils at a density of $1.04\,g\,cm^{-3}$ and room temperature with an angle of $60°$ between them. A $3\omega$ laser beam (comprised of $1.6 \times 10^4$ rays) with a 1 ns square pulse profile and 1 kJ of energy illuminates each of the two foils. The incidence angle and the SG8 phase plates (with very similar characteristics to the SG4 plates used in the experiment) determine the spot size and shape for each beam. The beams point at the center of the foils, albeit one of the beams is offset by $\sim 100\,\mu m$ in the Z direction towards the target's base to introduce an asymmetry that excites the $m = 1$ (kink) mode. Conversely, we introduce a small-amplitude, time-dependent, sinusoidal perturbation[13,42] on the transverse velocity components in the interaction region where the jet is formed, so as to excite the $m = 0$ (sausage) mode. The amplitude of the perturbation is 1% of the flow speed, with a period $= 0.1 \times \tau = 0.1$ ns, where $\tau$ is the system's timescale. The evolution of the system is followed for 5 ns.

**Data availability.** The authors declare that the data supporting the findings of this study are available within the article and its Supplementary Information files, and are available from the authors on request. The FLASH code is publicly available through the webpage of Flash Center, University of Chicago (flash.uchicago.edu).

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

## Acknowledgements

We thank the OMEGA operations and target fabrication crews for their assistance in carrying out these experiments and R. Frankel, and E. Doeg for their help in processing of CR-39 data used in this work. The experiments were supported in part by US DOE (Grant No. DE-FG03-09NA29553, No.DE-SC0007168), LLE (No.414090-G), NLUF (No.DE-NA0000877), FSC (No.415023-G) and LLNL (No. B580243). Numerical simulations were supported in part by the US DOE NNSA ASC under Field Work Proposal No. 57789 to the Argonne National Laboratory, and by NIH through resources provided by the Computation Institute and the Biological Sciences Division of the University of Chicago and Argonne National Laboratory (Grant 1S10OD018495-01). Partial support from the European Research Council under the European Community's Seventh Framework Programme (FP7/2007-2013)/ERC grant agreements no. 256973 is acknowledged. The software used in this work was developed in part by the DOE NNSA ASC- and DOE Office of Science ASCR-supported Flash Center for Computational Science at the University of Chicago. This research used resources from the Director's Discretionary Program of the Argonne Leadership Computing Facility, supported by DOE Office of Science User Facility (DE-AC02-06CH11357).

## Author contributions

C.K.L. conceived and led the experiments, and analyzed the data. P.T. and D.L. performed the FLASH numerical simulations, and contributed to data interpretation. M.J.R. contributed to execution and discussion of experiments. R.K.F. and D.H.F. supported the 4ω Thomson scattering measurements. F.H.S. and R.D.P. contributed to the development of proton radiography and the discussion of experiments. M.K., J.A.F., H.G.R., H.S., A.B.Z., P.A.A., H.S.P., B.A.R., D.D.R., S.C.W., R.B., A.F., S.X.H., T.C.S., P.H., C.C.K., R.P.D., G.G., P.A.N., S.V.L. and N.C.W. contributed to support the experiments and technical discussions. C.K.L., P.T., G.G. and D.L. wrote the paper.

## Additional information

**Competing financial interests:** The authors declare no competing financial interests.

