## [Peer Review File · Nature Communications]

Reviewers' comments:

Reviewer #1 (Remarks to the Author):

Referee Report for Manuscript #88442

The manuscript reports research efforts on the study of magnetized jets, and specifically regarding observations of the jet in the Crab Nebula. The authors present evidence of magnetically-driven instabilities forming in laser driven-jet stricture, and argue that the physics dominating here is similar to that in the astrophysical system, in that the jet structure and velocity history is driven primarily by MHD instabilities due to entrained magnetic field in the propagating jet.

The dimensionless scaling parameters are all in useful regimes, and in particular, I believe this is the highest magnetic Reynolds number achieved in jet experiments to date. This is a significant advance. The application of proton probing demonstrates the presence of electromagnetic fields in the jet and a structure consistent with B-field entrainment from the launch region as seen in simulation work. One point here is that in Table 1, the jet velocity is listed as ~ 400 km/s, but in the text this is quoted as 1000-1500 km/s from Thomson Scattering measurements (e.g. Page 9 & Fig 5). Why was the figure of 400 km/s used in the table and which values were used to calculate the dimensionless variables?

Overall, the authors make a convincing case that the development of structure in the experimental jets is driven by MHD modes due to the entrained magnetic field. This work I think also therefore presents the first direct measurements of entrained B-field inside the body of the jet in such experiments. The experimental work is supported by realistic analytical and detailed numerical calculations. In particular, the agreement of the experimental velocities, density and temperature with the simulation work over relatively large spatial extents is excellent.

It is not clear that the development of MHD instabilities in the jet body cause a change in the global directionality. In both the proton probing images and the simulation work, it appears that the main mass in the jet maintains a velocity along the propagation axis. The authors suggest that the jet is 'serpentine' from evidence of the magnetic field structure, but then show that the proton diagnostic is insensitive to mass density, and therefore not a direct measure of the flow direction. Are there Thomson scattering measurements available showing radial velocity components in the flow at certain regions from a helical $M=1$ structure? Is the plasma beta here sufficiently high that the magnetic pressure entirely determines the mass distribution throughout the jet body? Large deviations from original jet direction, as observed in the Crab Nebula jet, are not seen in the experiments even though estimated instability timescales seem appropriate in both cases. It seems likely that MHD instabilities are the driving force in both cases, but feel that this work has not proven this as unequivocally as they suggest.

In summary, the manuscript presents thorough, high quality research, and many researchers in the plasma physics and likely observational astronomy fields will be greatly interested. There are significant advances in the experimental parameter space attained, which make experiments more relevant to the astrophysical case than has been possible previously. However, a fully convincing link from the laboratory to the astrophysical is extremely difficult and I feel has not been demonstrated here. I would be interested to see authors' comments, and whilst I would not recommend publication at this time clarifications may help support this in a revised manuscript.

Reviewer #2 (Remarks to the Author):

Here are some comments.

(1) They produce a proton jet in the laboratory by laser-foil interaction. I am not clear why the

mobile electrons are not extracted from the foil by laser irradiation.

(2) In Figure~1 c, what does the vertical axis represent? The proton ($\sim 1-10\text{MeV}$) speed is much shorter than the speed of light (Lorentz factor ~ 1), while the flow in the Crab nebula will be $v > c/3$, for which the special relativistic effect might be important. (Mignone, A. et al. 2013).

(3) In Figure~3, it is not clear to me how the outflow from the flat foil by irradiation laser can create the collimated jet. Maybe, some explanation could help me. Does the angle of the two foil affect to only $|B_{\phi}/B_p|$? Moreover, if magnetic field around jet is so small, why is the jet collimated?

(4) What does the actual value of $|B_{\phi}/B_p|$ in the experiment? Comparing the initial magnetic field configuration between figure 1 and figure 3, I though after reconnection $|B_{\phi}/B_p|$ is smaller for Figure 1, in which the initial magnetic field has poloidal component, while no poloidal component in Figure 3.

(5) In there any good reason we can compare the proton jet in the laboratory with the MHD jet in the simulation? I think they compare the different simulations. The Crab nebula also the jet should be MHD flow.

(6) In Table 1, the scaled speed to the Crab jet exceeds the speed of light, which may not be reasonable. I understand it is just discussion of order of magnitude. However, choosing the scale factors, if it is arbitrary, that limit the flow speed below speed of light may be more appropriate.

(7) Since their simulation is radiative MHD, what is the time scale of the radiate cooling, which will affect the magnetization parameters? Is this consistent with the Crab nebula?

Response to Reviewer's comments

“Scaled laboratory experiments explain the kink behavior of the Crab Nebula jet”

By C. K. Li, P. Tzeferacos, D. Lamb, G. Gregori et al.

We thank the Reviewers for their careful reading of our manuscript and their insightful comments. We appreciate the first Reviewer's acknowledgment of the value and interest of our work -- the first systematic, scaled laboratory experiments using laser-driven plasma jets for studying the fascinating kink behavior of the Crab Nebula jet. The Reviewers' constructive comments, largely requested for technical clarity, bear on important aspects of the new physical insights we are presenting. They have helped us write a stronger article that places the uniqueness and importance of our studies in a broader context that can be more fully appreciated by the readers/researchers.

We have made modifications to the paper to emphasize more clearly the importance and novelty of our work. We have added information and discussions to the text. Most importantly, the implications of this scaled laboratory experiment for the Crab Nebula jet have been substantially strengthened. We feel that this work not only fills a critical gap in the experiments for addressing the kink behavior of the Crab jet but also opens up an opportunity for studying astrophysical phenomena in laboratory.

Each of the points made by the Reviewers is either addressed in the revised manuscript, and noted here, or addressed in detailed responses to the Reviewers. In the following itemized discussion we present the Reviewer's comments in (blue) italics; our reply is in (black) plain type; and modifications in the text (listed in this discussion) are in red.

Response to Reviewer #1:

(1) “The manuscript reports research efforts on the study of magnetized jets, and specifically regarding observations of the jet in the Crab Nebula. The authors present evidence of magnetically-driven instabilities forming in laser driven-jet structures, and argue that the physics dominating here is similar to that in the astrophysical system, in that the jet structure and velocity history is driven primarily by MHD instabilities due to entrained magnetic field in the propagating jet.

The dimensionless scaling parameters are all in useful regimes, and in particular, I believe this is the highest magnetic Reynolds number achieved in jet experiments to date. This is a significant advance. The application of proton probing demonstrates the presence of

electromagnetic fields in the jet and a structure consistent with B-field entrainment from the launch region as seen in simulation work. One point here is that in Table 1, the jet velocity is listed as ~ 400 km/s, but in the text this is quoted as 1000-1500 km/s from Thomson Scattering measurements (e.g. Page 9 & Fig 5). Why was the figure of 400 km/s used in the table and which values were used to calculate the dimensionless variables?"

We thank the Reviewer for his/her encouraging comments. With regard to the jet velocity, the value $V_j \sim 1500$ km/s we give in the text (measured with Thomson scattering and proton radiography and simulated with the FLASH radiative-hydrodynamic code, as shown in Fig. 5a) is the flow velocity close to the jet head; the value $V_j \sim 400$ km/s listed in Table 1 is the flow velocity of the plasma jet in the jet-launching region. As indicated in the footnotes to Table 1, all of the values we used to calculate the dimensionless variables listed in Table 1 come from the region where the jet is formed.

(2) "Overall, the authors make a convincing case that the development of structure in the experimental jets is driven by MHD modes due to the entrained magnetic field. This work I think also therefore presents the first direct measurements of entrained B-field inside the body of the jet in such experiments. The experimental work is supported by realistic analytical and detailed numerical calculations. In particular, the agreement of the experimental velocities, density and temperature with the simulation work over relatively large spatial extents is excellent."

Again, we thank Reviewer for his/her kind comments.

(3) "It is not clear that the development of MHD instabilities in the jet body cause a change in the global directionality. In both the proton probing images and the simulation work, it appears that the main mass in the jet maintains a velocity along the propagation axis. The authors suggest that the jet is 'serpentine' from evidence of the magnetic field structure, but then show that the proton diagnostic is insensitive to mass density, and therefore not a direct measure of the flow direction."

We appreciate the Reviewer's insightful comment. As the Reviewer correctly points out, our conclusion that the jet is "serpentine" is largely based on the measured magnetic field structure, since proton radiography is sensitive to magnetic fields but not to the plasma density. In this experiment, the plasma density was low ($\sim 10^{18}$ - 10^{19} /cm³), and so was transparent to the backlighting protons. While we do not have a direct experimental measurement of density modulation, the kinked shape of the jet can be inferred from the fact that the magnetic field threads the plasma of the jet and is "frozen into" the plasma flow. This is true when non-ideal effects such as magnetic diffusion in the generalized Ohm's law are small and the self-generated magnetic fields are advected with the flow, as described by Faraday's equation. The magnetic Reynolds number is large enough in the experiment for this condition to hold. Consequently, the

plasma in the jet must follow the field topology and its evolution, which is locally kinked but globally “collimated” along the propagation axis.

To clarify this physics picture, we have added a short paragraph in the text (which is now the second paragraph on page 4) that reads:

“The magnetic Reynolds number in the experiment is large enough for the magnetic field to be “frozen into” the plasma flow. Consequently, the plasma in the jet must follow the field topology and its evolution, which is locally kinked but globally “collimated’ along the propagation axis.”

We tried to measure the jet plasma flow directly with x-ray imaging. However, limited by the total driving energy on OMEGA laser, we had to use low-Z material (plastic) for the target in order to obtain high jet flow velocities, and therefore high magnetic Reynolds numbers, given the limit imposed by the total energy of the OMEGA laser. X-ray imaging of the low-Z and low-density jet plasma was unsuccessful. We plan to extend these experiments to the National Ignition Facility (NIF), which has a much larger laser energy that can drive higher Z targets to high velocities and achieve higher densities, both of which make imaging the plasma jet easier.

(4) “Are there Thomson scattering measurements available showing radial velocity components in the flow at certain regions from a helical $M=1$ structure?”

The configuration of the Thomson-scattering diagnostic in the experiment only allowed measurements of the flow velocity along the propagation axis. However, the helical structure can be inferred from the proton radiography features using the flux-freezing argument mentioned above, and from the validated FLASH simulations.

(5) “Is the plasma beta here sufficiently high that the magnetic pressure entirely determines the mass distribution throughout the jet body? “

In the experiment, colliding plumes create a modulated central “spine” (backbone) region with strong magnetic fields that is surrounded by asymmetrically-distributed weaker fields, as discussed in the text and Fig 4a-4c. If the field is sufficiently large and has a non-uniform toroidal component B_ϕ , current-driven MHD kink modes are excited with the susceptibility of this happening increasing with increasing $|B_\phi/B_p|$ (see Fig. 4b). The image in Fig. 4c shows the values of $\beta = 8\pi nkT/B^2$ (the ratio of plasma thermal to magnetic pressures) in the validated FLASH simulations. In the jet core, $\beta < \sim 1$, which indicates that the flow is magnetically dominated; however, in the surrounding plasma, however, $\beta > 1$, which indicates that the jet is globally collimated due to inertial confinement and magnetic tension, but locally kinked.

(6) “Large deviations from original jet direction, as observed in the Crab Nebula jet, are not seen in the experiments even though estimated instability timescales seem appropriate in both cases. It seems likely that MHD instabilities are the driving force in both cases, but feel that this work has not proven this as unequivocally as they suggest.”

We appreciate the reviewer’s comment about the large deviations from the original jet direction that are observed in the Crab Nebula jet. The aim of our work is to demonstrate through a scaled laboratory experiment that the instability observed in the Crab Nebula jet is a result of the kink instability, as posited and simulated by many theoretical astrophysicists. The experimental data and the validated three-dimensional FLASH MHD simulations of the experiment indicate that, for the physical properties of the laboratory experiment, the plasma jet is unstable to the kink instability with a growth rate that, when scaled to the physical properties of the Crab Nebula jet, can explain the similar behavior observed in it. (That is, the kinks along the jet that are visible in a close examination of the November 6, 2008, and May 11, 2011, Chandra x-ray images of the jet shown in Supplementary Figure 1; see also Figure 14 in ref. 13 and the movies linked to it).

The laboratory experiment was not designed to explain the large deviation from the original jet direction (i.e., the “fish hook” shape associated with the beginning of the backflow) in the Crab Nebula jet. This feature is thought to be due to the combination of the kink instability and the backpressure resulting from the fact that the jet is moving through an overdense plasma. The plasma jet in the laboratory experiment is moving through vacuum, and so does not experience a similar backpressure. The jet body however exhibits kinks, knots, and radial deflections that mimic the morphology observed in Chandra’s X-ray imaging of the Crab jet. This becomes readily apparent when considering the simulated current density map of the flow, which has been added as Figure 6b in the manuscript. Such a picture of a “current-carrying” jet that is potentially unstable to current-driven modes is in agreement with a number of investigators that model pulsar wind nebulae numerically (see for instance ref. 10, 13). To highlight the morphological similarities between the laboratory jet and the astrophysical flow we have added the following text in the Discussion section (first paragraph) and a new panel in Fig. 6.

“The morphological similarities between the Crab jet (Supplementary Figure 1) and the laboratory jet can be clearly seen in the simulated current density map (Figure 6b). The latter reveals kinks, knots, and large-scale radial deflections that are reminiscent of the structures and dynamics observed in the Crab pulsar outflow. This picture of a “current-carrying” jet is in agreement with existing numerical efforts on modeling the Crab jet^{10,13} and the morphology mimics the jet structures observed in Chandra X-ray imaging¹.”

(7) *“In summary, the manuscript presents thorough, high quality research, and many researchers in the plasma physics and likely observational astronomy fields will be greatly interested. There are significant advances in the experimental parameter space attained, which make experiments more relevant to the astrophysical case than has been possible previously. However, a fully convincing link from the laboratory to the astrophysical is extremely difficult and I feel has not been demonstrated here. I would be interested to see authors' comments, and whilst I would not recommend publication at this time clarifications may help support this in a revised manuscript.”*

We appreciate the Reviewer’s careful reading of the manuscript and insightful comments. We have made a number of modifications to the text that strengthen the connection between the laboratory experiment and the astrophysical system, and have added text to address the Reviewer’s concerns.

Response to Reviewer #2:

(1) *“They produce a proton jet in the laboratory by laser-foil interaction. I am not clear why the mobile electrons are not extracted from the foil by laser irradiation”.*

We believe that there may be some confusion about the nature of the laser-produced jet. It is not a proton jet but a plasma jet, comprised of ions (carbon and hydrogen) and electrons. We describe this in the text in the Methods section. To clarify this, we have added the following phrase in the caption of Fig. 1:

“.....and resulting plasma jets comprised of ions and electrons.....”

(2) *“In Figure~1 c, what does the vertical axis represent? The proton (~1-10MeV) speed is much shorter than the speed of light (Lorentz factor~1), while the flow in the Crab nebula will be $v>c/3$, for which the special relativistic effect might be important. (Mignone, A. et al. 2013)”.*

In Figure 1c, the unit of the vertical axis is proton counts, which is proportional to the proton fluence. The lineouts we plot in this figure show the modulation of the proton counts along the direction of the plasma jet, which enables us to measure the wavelength of the jet modulation. To clarify this, we have added a phrase in the caption of Fig. 1c which reads: *“.....The unit of the vertical axis is proton counts, which is proportional to the proton fluence.....”*.

We compare the velocities of the plasma jet with the Crab Nebula jet using the Lorentz factor and scaled them based on hydrodynamic similarity. The flow velocity of the Crab Nebula jet is $V_j \sim 0.4c$, which corresponds to a Lorentz factor

$$\gamma = \frac{1}{\sqrt{1 - \left(\frac{V_j}{c}\right)^2}} \approx 1.09;$$

i.e., a Lorentz factor of about unity. This indicates that relativistic effects are modest in the Crab Nebula jet, allowing us to scale the velocity of the laboratory plasma jet to the Crab Nebula jet using the approximation of non-relativistic hydrodynamic scaling (see Eq. 3 in the manuscript). To clarify this, we now explicitly give the Lorentz factor for the Crab jet in the third paragraph of the Discussion section, page 7: *“.....($\Gamma = 1$ for the laboratory plasma jet and $\Gamma \approx 1.09$ for the Crab Nebula jet³⁰)......”*

(3) *“In Figure~3, it is not clear to me how the outflow from the flat foil by irradiation laser can create the collimated jet. Maybe, some explanation could help me. Does the angle of the two foil affect to only $|B_\phi/B_p|$? Moreover, if magnetic field around jet is so small, why is the jet collimated?”*

As described in our point (1), the collision of the two laser-driven plasma plumes will form a supersonic plasma jet in the midplane, with high velocities and associated self-generated magnetic fields. The mechanism for generating the plasma jet shown in Fig. 3 is the same as that shown in Fig. 1, that is, the collision of two plasma plumes. As the Reviewer correctly points out, the angle between the two foils affects the relative ratio of B_ϕ/B_p . It should be noted, however, that this angle is not the only experimental parameter that affects this ratio: the laser intensity, the duration of the laser pulse and the size of the laser-beam spots also affect the ratio.

In our experiments, we purposely designed a jet that has a large B_ϕ/B_p ratio in the jet core region ($B_\phi/B_p > \sim 1$, see Fig. 4b) and is therefore unstable to MHD kink instability (Fig. 1a), using two CH foil targets at a 60° angle. On the other hand, the design using a flat foil target was chosen to minimize B_ϕ around the jet, so as to obtain a stable, collimated plasma jet (Fig. 3).

The jet shown in Fig. 3 is predominately collimated by inertial confinement due to the hydrodynamic compression (collision) of the two high β plasma plumes. The effects of magnetic confinement are minimal due to the small B_ϕ around the jet.

To clarify this, we have added a sentence in the end of caption of Fig. 3: “.....The jet is predominately collimated by inertial confinement due to the hydrodynamic compression produced by the collision of the two plumes.”

(4) “What does the actual value of $|B_\phi|/B_p$ in the experiment? Comparing the initial magnetic field configuration between figure 1 and figure 3, I though after reconnection $|B_\phi|/B_p$ is smaller for Figure 1, in which the initial magnetic field has poloidal component, while no poloidal component in Figure 3.”

Experimentally, it is extremely difficult to measure the ratio B_ϕ/B_p experimentally using side-on proton radiography. Our arguments therefore rely on the other experimental measurements we made and the nature of MHD current-driven instabilities, together with the validated FLASH MHD simulations. In particular,

1. In Fig. 1, the jet modulation with dominant MHD unstable modes $m = 0$ (sausage) and $m = 1$ (kink) indicates $B_\phi/B_p > 1$.
2. In Fig. 3, the jet propagation is stable, with the backlighting protons predominantly deflected from the upper half of the jet to the lower half, indicating that $B_p \gg B_\phi$ suppressing current-driven instabilities.

To clarify this, we have explicitly added a phrase in caption of Fig. 3: “...; i.e., $B_\phi/B_p < 1$...”

(5) “In there any good reason we can compare the proton jet in the laboratory with the MHD jet in the simulation? I think they compare the different simulations. The Crab nebula also the jet should be MHD flow.”

As we clarified in our response to comment (1), the comparisons we perform are among a laser-driven, laboratory plasma jet that is in the MHD regime, an MHD simulation of the laboratory plasma jet, and – using appropriate MHD scaling – the MHD Crab Nebula jet.

(6) “In Table 1, the scaled speed to the Crab jet exceeds the speed of light, which may not be reasonable. I understand it is just discussion of order of magnitude. However, choosing the scale factors, if it is arbitrary, that limit the flow speed below speed of light may be more appropriate.”

We agree with the Reviewer’s point and appreciate his/her understanding. Indeed, our scaling of the laboratory experiments to the Crab Nebula jet uses a nonrelativistic hydrodynamic scaling (as we discussed in our response to the Reviewer’s second comment), with the intent to present an approximate scaling that highlights the similarity in underlying physics between the laboratory experiment and the astrophysical system. As the Reviewer points out, the scaled velocity should be less than the speed of light. As the scaling is approximate, we now give a value for the scaled velocity that is less than the speed of light (scaled $V_j < 3 \times 10^5$ km/s), and make the corresponding change in Table 1.

(7) “Since their simulation is radiative MHD, what is the time scale of the radiate cooling, which will affect the magnetization parameters? Is this consistent with the Crab nebula?”

In these experiments, the plasma jets were generated with laser-irradiating plastic (CH) foils that have low values of Z (average $Z \sim 3.5$) and the jet propagation has been an adiabatic rarefaction. The effects of plasma radiation cooling were relatively small and negligible, consistent with the case of the Crab Nebula jet. To quantitatively confirm this, we have evaluated and compared the Radiation number [\mathcal{R} , from ref.19 in the manuscript], a dimensionless number defined as the ratio of material energy flux to the radiative energy flux, for both the laboratory system and the Crab jet. The estimated Radiation number \mathcal{R} is $\sim 3 \times 10^5$ for the laboratory system and 1×10^{18} for the Crab jet, which are both adequately large, indicating the radiation cooling does not affect the jet dynamics for both systems. In order to clarify this point, we have added the calculated Radiation numbers in Table 1. We point out, however, that the laboratory plasma jet is “cold” (in the sense that the thermal velocity of the ions is much smaller than the bulk velocity of the jet), due to adiabatic cooling of the flow. The Crab Nebula jet is also cold.

TABLE I. Physical parameters and similarity scaling between the laboratory jet and the Crab nebula jet

Parameters and scales		Plasma jet in OMEGA experiment [†]	Scaled to the Crab nebula [‡]	The kinked jet in the Crab nebula [‡]
Temperature	T_e	~ 300 eV		~ 1 eV
Ionization state	Z	~ 3.5		~ 1
Number density	n_e	$\sim 5 \times 10^{19}$ cm ⁻³		$\sim 10^{-2}$ cm ⁻³
Pressure	P	$\sim 4 \times 10^5$ bar		$\sim 4 \times 10^{-14}$ bar
Jet radius	r_j	$\sim 5 \times 10^{-2}$ cm		~ 1 pc
Jet velocity	v_j	~ 400 km s ⁻¹	$< 3 \times 10^5$ km s ⁻¹	$\sim 1.2 \times 10^5$ km s ⁻¹
Time scale	τ	$\sim 10^{-9}$ s	~ 1.5 Yrs	\sim few Yrs
Magnetic field	B	~ 2 MG	~ 0.6 mG	~ 1 mG
Thermal plasma beta	β	$\sim 0.1-1$		$\ll 1$
Magnetization parameter	σ	$\sim 1-6$		≥ 1
Mach number	M	~ 3		$\gg 1$
Reynolds number	Re	$\sim 2 \times 10^3$		$\sim 2 \times 10^{17}$
Péclet number	Pe	$\sim 1-5$		$\sim 4 \times 10^{15}$
Magnetic Reynolds number	Re_M	$\sim 3 \times 10^3$		$\sim 1 \times 10^{22}$
Biermann number	Bi	~ 6		$\sim 6 \times 10^8$
Radiation number	Π	$\sim 3 \times 10^5$		$\sim 1 \times 10^{18}$

[†] Near the region of jet launching.

[‡] Near the region of the pulsar pole.

We thank the Reviewer for the insightful and detailed comments.

Reviewers' comments:

Reviewer #1 (Remarks to the Author):

The authors have addressed the issues raised, and publication is now appropriate

Reviewer #2 (Remarks to the Author):

I am satisfied with their replies, except for (7). In (7), they answered that the radiation cooling effect can be ignored for the Crab pulsar and they also said the Crab jet is cold.

(a) In the manuscript, the authors compare their interesting laboratory jets with Chandra images of the Crab jets. It is usually discussed that the cooling process of the Crab nebula is the non-thermal radiation. For example, the X-ray emissions from the Crab jet is produced by the synchrotron radiation of the shocked pulsar wind particles. We can estimate that the cooling time scale of the synchrotron radiation of the particles that emits X-rays with $B \sim 1\text{mG}$ is of order of ~ 4 years, which is of order of the crossing time scale over the jet.

I checked the paper of Cross et al. (2014), in which I think they discussed the cooling with the thermal radiation. Is it still applicable to the Crab nebula jet?

(b) Authors also replied that the Crab jet is "cold". It is not clear to me why we can say the "cold". For the Crab nebula, the shocked particles are almost speed of light, which is larger than bulk velocity. Moreover, the internal energy is much larger than the kinetic energy of the bulk flow. So I would say the Crab jet is "hot", in the sense that internal energy is much larger than kinetic energy of the bulk flow.

It is not clear to me how author estimated the temperature of the Crab jet ($\sim 1\text{eV}$ in Table1).

Response to the further comments from Reviewer #2
“Scaled laboratory experiments explain the kink behavior of the Crab Nebula jet”

By C. K. Li, P. Tzeferacos, D. Lamb, G. Gregori et al.

We thank again the insightful comments from Reviewer #2. In the following itemized discussion we present the Reviewer’s comments in (blue) italics; our reply is in (black) plain type; and modifications in the text (listed in this discussion) are in red.

I am satisfied with their replies, except for (7). In (7), they answered that the radiation cooling effect can be ignored for the Crab pulsar and they also said the Crab jet is cold.

We are glad to clarify further our understanding of the nature of the cooling in the Crab Nebula jet, and the reasons why we said the jet is cold.

(a) In the manuscript, the authors compare their interesting laboratory jets with Chandra images of the Crab jets. It is usually discussed that the cooling process of the Crab nebula is the non-thermal radiation. For example, the X-ray emissions from the Crab jet is produced by the synchrotron radiation of the shocked pulsar wind particles. We can estimate that the cooling time scale of the synchrotron radiation of the particles that emits X-rays with $B \sim 1mG$ is of order of ~ 4 years, which is of order of the crossing time scale over the jet. I checked the paper of Cross et al. (2014), in which I think they discussed the cooling with the thermal radiation. Is it still applicable to the Crab nebula jet?

The reviewer is correct that the radiation parameter Π we quote in the paper refers to the cooling due to thermal bremsstrahlung. This process is often the dominant cooling process in laboratory laser experiments. We therefore presented the value of this parameter to show that this process can be neglected in our laboratory experiments as well as in the Crab Nebula jet.

The reviewer is correct that non-thermal synchrotron emission is the dominant radiative cooling process in the Crab Nebula jet and that the cooling timescale (see, e.g., Komissarov & Lyubarsky, 2004) is comparable to the travel time along the length of the jet. However, the radiated power is estimated to be only $\sim 10\%$ of the instantaneous rate of energy input from the pulsar (see, e.g., Shibata et al. 2003; Del Zanna et al. 2006; Porth et al. 2013). This justifies our assumption that adiabatic cooling is the dominant cooling process in the jet, an assumption other investigators have made before us (see, e.g., Del Zanna et al. 2006).

(b) Authors also replied that the Crab jet is "cold". It is not clear to me why we can say the "cold". For the Crab nebula, the shocked particles are almost speed of light, which is

larger than bulk velocity. Moreover, the internal energy is much larger than the kinetic energy of the bulk flow. So I would say the Crab jet is "hot", in the sense that internal energy is much larger than kinetic energy of the bulk flow. It is not clear to me how author estimated the temperature of the Crab jet (~1eV in Table 1).

Enthalpy $w = \rho c^2 + [\Gamma p / (\Gamma - 1)]$ is the quantity of interest in the mildly relativistic plasma in the Crab Nebula jet. The reviewer is correct that this quantity is large, and in this sense, one might say the jet is “relativistically hot”. In the nebula, w is dominated by the thermal pressure p , but in the pulsar wind, w is dominated by the kinetic energy ρc^2 due to the bulk flow velocity (Del Zanna et al. 2006). Thus the plasma in the jet is “cold,” in the sense that the thermal pressure $p < \rho c^2$ (Del Zanna et al. 2004).

This potential confusion can be avoided by discussing the Crab Nebula jet in terms of its energy budget. As mentioned in the article, we compare the experiments with the properties of the Crab Nebula jet at its base (i.e., close to the pulsar pole and away from the termination shock where the flow becomes considerably thermalized). At the base of the jet, the kink-unstable spine is Poynting-flux dominated and $E_{\text{magnetic}} > E_{\text{kinetic}} > E_{\text{thermal}}$ (see also cases A2 and A3 in the top panel of Fig. 18 in Mignone et al. 2013, which better match the characteristics of the Crab Nebula jet). This is also the case in our laboratory jet.

We do not discuss the energy budget of the Crab Nebula jet in the article, since this information does not contribute to the exposition of the nature of the current driven instabilities. However, we hope that what we have written here clarifies sufficiently what we wrote in our previous response.

Lastly, the 1 eV temperature we quoted in the table is a lower limit on the jet temperature that is in line with the temperatures of the filaments (Fesen & Kirshner, 1982). Becker & Aschenbach (1995) calculated an upper limit on the jet temperature from ROSAT data, estimating temperatures < 130 eV at the pulsar surface (i.e., close to the jet launching region). We have updated the Table 1 to show the full temperature range (1 – 130 eV) consistent with these results.

REVIEWERS' COMMENTS:

Reviewer #2 (Remarks to the Author):

I can accept their reply and the current manuscript is appropriate for the publications.